# Precipitation Projection in Cambodia Using Statistically Downscaled CMIP6 Models

Seyhakreaksmey Duong [1], Layheang Song [1,2] and Rattana Chhin [1,2,*]

1 Faculty of Hydrology and Water Resources Engineering, Institute of Technology of Cambodia, Russian Federation Blvd., Phnom Penh P.O. Box 86, Cambodia; duong_seyhakreaksmey@dtc2.itc.edu.kh (S.D.); layheang.song@itc.edu.kh (L.S.)
2 Research and Innovation Center, Institute of Technology of Cambodia, Russian Federation Blvd., Phnom Penh P.O. Box 86, Cambodia
* Correspondence: rattana@ric.itc.edu.kh

**Abstract:** The consequences of climate change are arising in the form of many types of natural disasters, such as flooding, drought, and tropical cyclones. Responding to climate change is a long horizontal run action that requires adaptation and mitigation strategies. Hence, future climate information is essential for developing effective strategies. This study explored the applicability of a statistical downscaling method, Bias-Corrected Spatial Disaggregation (BCSD), in downscaling climate models from the Coupled Model Intercomparison Project Phase 6 (CMIP6) and then applied the downscaled data to project the future condition of precipitation pattern and extreme events in Cambodia. We calculated four climate change indicators, namely mean precipitation changes, consecutive dry days (CDD), consecutive wet days (CWD), and maximum one-day precipitation (rx1day) under two shared socioeconomic pathways (SSPs) scenarios, which are SSP245 and SSP585. The results indicated the satisfactory performance of the BCSD method in capturing the spatial feature of orographic precipitation in Cambodia. The analysis of downscaled CMIP6 models shows that the mean precipitation in Cambodia increases during the wet season and slightly decreases in the dry season, and thus, there is a slight increase in annual rainfall. The projection of extreme climate indices shows that the CDD would likely increase under both climate change scenarios, indicating the potential threat of dry spells or drought events in Cambodia. In addition, CWD would likely increase under the SSP245 scenario and strongly decrease in the eastern part of the country under the SSP585 scenario, which inferred that the wet spell would have happened under the moderate scenario of climate change, but it would be the opposite under the SSP585 scenario. Moreover, rx1day would likely increase over most parts of Cambodia, especially under the SSP585 scenario at the end of the century. This can be inferred as a potential threat to extreme rainfall triggering flood events in the country due to climate change.

**Keywords:** Coupled Model Intercomparison Project Phase 6; Bias-Corrected Spatial Disaggregation; consecutive dry days; consecutive wet days; maximum 1-day precipitation



## 1. Introduction

Climate change would likely impact ecosystems, the hydrologic cycle, and human lifestyles. Several sectors of the human system, such as the economy, agriculture, transportation, health, and water resources, are also vulnerable to these impacts. Many studies have examined the consequences of extreme weather events and natural disasters, but most of these assessments have not considered long-term changes in climate patterns or long-term trends in the intensity or frequency of extreme events [1]. The hydrologic cycle is changing due to global warming, leading to increasingly severe weather events. The modifications to the hydrologic cycle are causing a variety of health issues, such as respiratory issues, waterborne infections, and ailments brought on by heat [2]. Furthermore, climate change poses a major threat to the agricultural sector by altering crop resistance through changing

climate patterns and surface temperatures. Climate change adaptation in this sector will require significant effort and resources, but it is essential to ensure food security in the future [3]. Moreover, transportation is essential for economic growth but is also a major source of greenhouse gas emissions. Efforts to reduce emissions from transportation, such as promoting electric vehicles and alternative fuels, could reduce transportation efficiency and slow down economic growth [4]. On top of that, climate change is expected to cause more frequent and severe floods, which are a major threat to road infrastructure in urban areas. This is because urbanization has led to increased impervious surfaces, which cannot absorb water, and decreased green spaces, which can help to slow down and reduce heavy rainfall [5]. Consequently, temperature and precipitation alteration would affect developing countries heavily due to their limited adaptation capacity; the severity depends on the location of the regions [6].

Climate projection using original coarse-resolution climate model over a small domain is not enough to investigate detailed local climate change features. Therefore, a downscale of the climate model is essential before using the climate model for future climate projection. Climate model downscaling involves refining global climate model projections by incorporating local-level factors like topography, vegetation, and land use. This process enhances the accuracy and regional relevance of climate projections [7]. Two main types of downscaling methods can be used to study local-scale climate change; it can be either statistical downscaling or dynamical downscaling [8,9]. Statistical downscaling uses statistical relationships between large-scale climate variables and local weather data to project how climate change will affect a specific region. This method is less computationally expensive than dynamical downscaling. Whereas, the dynamical downscaling uses the output from Global Climate Models (GCMs) as boundary conditions to drive the regional climate model to derive smaller-scale information [10]. This method would be more expensive and require high-skill human resources to perform the work.

There are a few previous research studies about climate change projection in Cambodia. A study on temperature and rainfall projection in Cambodia was done using the Coupled Model Intercomparison Project Phase 3 (CMIP3) climate model under two emission scenarios [11]. Additionally, a World Bank report projected the future climate of Cambodia by utilizing the CMIP5 climate model under four climate change scenarios to foresee changes in temperature and rainfall [12]. Similarly, research on climate change in one of Cambodia's provinces was also done using the CMIP5 climate model from three different Representative Concentration Pathways (RCPs) scenarios, using a downscaling method on precipitation and temperature [13]. However, the climate projection based on the newly released CMIP6 climate model specifically over Cambodia has not been carried out yet. Moreover, most of the previous studies focused only on the changes in mean precipitation; limited studies focus on the changes in extreme precipitation, which is very pronounced under the impact of climate change. Therefore, a study related to the projected changes in extreme precipitation in Cambodia based on CMIP6 models is required to fill in this gap.

Dealing with climate change impacts requires adaptation and mitigation strategy before the consequences occur. Therefore, the study of future climate information must be done to formulate effective strategies. In this study, we made climate change projections for the near future period (2015–2045) and far future period (2070–2100) by using Coupled Model Intercomparison Project Phase 6 (CMIP6) models. We calculated the changes in mean precipitation and extreme events, such as consecutive dry days (CDD), consecutive wet days (CWD), and maximum 1-day precipitation (rx1day). However, the coarse resolution of the CMIP6 climate model limits their application to small regions, like Cambodia, potentially producing high bias results. Hence, we explore the applicability of the Bias-Corrected Spatial Downscaling (BCSD) method for downscaling the CMIP6 models for investigating the climate change signal in Cambodia. This research aims to project future precipitation patterns by using CMIP6 climate model, combined with the BCSD method, while consuming fewer resources and less time. Thus, using the BCSD method is the key to

the cost-effective approach in the project since it is a statistical downscaling method that requires only a personal computer and climate data input.

## 2. Materials and Methods

### 2.1. Study Area

In this study, we selected the whole of Cambodia as the main scope for precipitation projection. Cambodia is a country located in the Southeast Asia region, sharing borders with Thailand, Vietnam, Laos, and the Gulf of Thailand. The country has a total population of 16.5 million (2019 census), with an area of 181,035 square kilometers. Cambodia's landscape is characterized by a central plain surrounded by hills and mountains (Figure 1). The plain is home to the Tonle Sap Lake, the largest freshwater lake in Southeast Asia. The uplands and low mountains of Cambodia are home to a variety of forests, including tropical rainforests, deciduous forests, and mangrove forests. The average annual rainfall in Cambodia ranges from 1400 mm to 4000 mm. However, some parts, such as Northeastern and Southwestern Cambodia, receive more rainfall than others [12]. Cambodia is under the influence of monsoon rainfall, which is divided into three periods: pre-monsoon, summer monsoon, and post-monsoon [14].

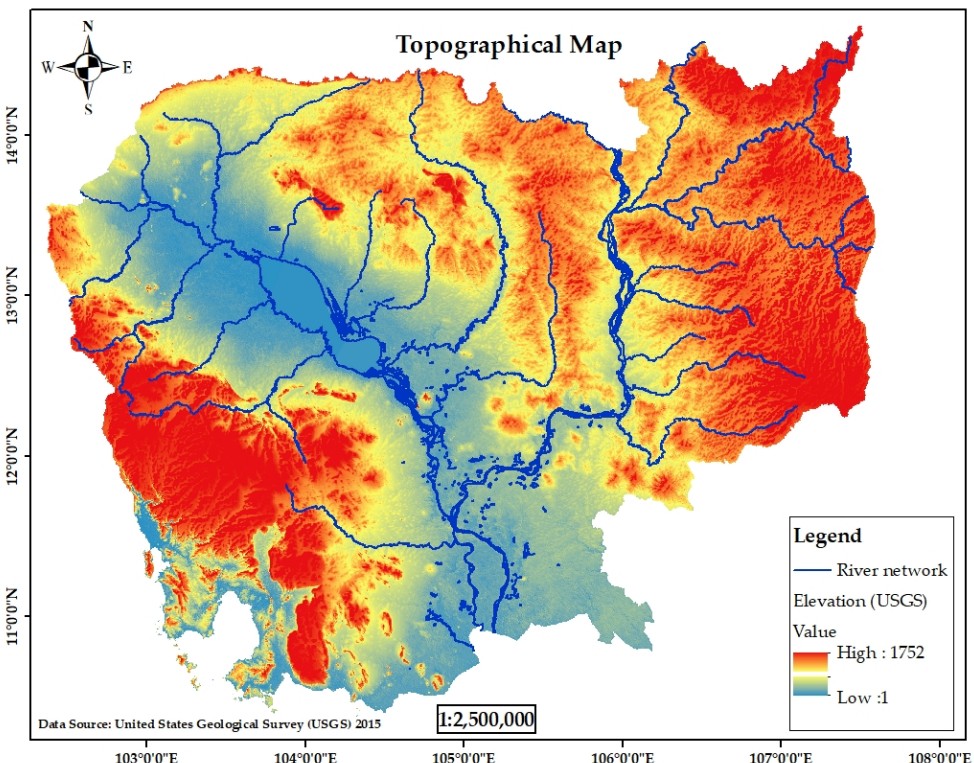

**Figure 1.** Elevation Profile of Cambodia.

### 2.2. Dataset

Statistical downscaling employs statistical relationships between observation and climate model to replicate a climate condition over a specific region in more detail. Therefore, two types of datasets were used in this research: observed data and climate model data (Table 1). We selected five climate models from CMIP6 [15] by choosing one model or one ensemble from each institution to ensure that they are independent from each other (avoid inter-model dependency). Those models are Euro-Mediterranean Centre for Climate Change (CMCC), Model for Interdisciplinary Research on Climate (MIROC), Max Planck Institute for Meteorology (MPI), Meteorological Research Institute (MRI), and Nanjing University of Information Science and Technology. In addition, the observed data were obtained from Multi-Source Weighted-Ensemble Precipitation (MSWEP) [16].

**Table 1.** Detailed information on the dataset.

| Data Center | Model Acronym | Resolution | Frequency | Available Period |
| --- | --- | --- | --- | --- |
| MSWEP | MSWEP | 0.1° | daily | 1979 to present |
| CMCC | CMCC-CM2-SR5 | 1.25° | daily | 1850–2100 |
| MIROC | MIROC6 | 1.4° | daily | 1850–2100 |
| MPI | MPI-ESM1-2-HR | 0.94° | daily | 1850–2100 |
| MRI | MRI-ESM2-0 | 1.13° | daily | 1850–2100 |
| NUIST | NUIST-NESM3 | 1.88° | daily | 1850–2100 |

*2.3. Bias-Corrected Spatial Disaggregation*

The climate model from CMIP6 has a relatively coarse resolution, which is not suitable for extreme precipitation analysis in the local area. Therefore, downscaling the climate model is essential before applying it to local-scale studies, as it helps to reduce large biases in the model output. In this study, we applied a statistical downscaling method called "Bias-Corrected Spatial Disaggregation" (BCSD), which was first applied by [17]. The process of BCSD method contains three main steps described in detail in [18]. The flowchart in Figure 2 shows the detailed process of the BCSD method. Step 1 is to interpolate the observation data to the same spatial scale as that of the climate model. Step 2 is to use the upscaled observation to bias-correct the climate model, using the quantile mapping method. Step 3 is to perform spatial disaggregation on the bias-corrected climate model to a finer resolution.

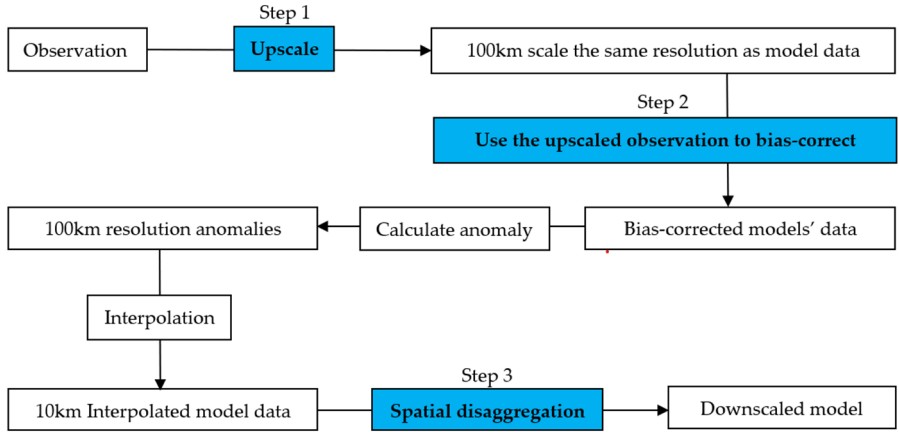

**Figure 2.** Flowchart describing the BCSD process.

2.3.1. Spatial Linear Interpolation of the Observed Data

The observed data were interpolated from its original scale to the same scale as the model data by using the bilinear interpolation method. Note that this process was done without any intervention of bias correction and spatial disaggregation. Afterward, the interpolated observed data were used to bias-correct the model data to remove as much of the model bias as possible.

2.3.2. Quantile Mapping

Quantile mapping was employed for bias correction. In other words, quantile mapping enhances the reliability of climate models by adjusting the distribution of their outputs to more closely match the observed distribution of climate variables. Quantile mapping was achieved by identifying a transfer function that captures the relationship between the observed and climate model data. Once the transfer function is determined, it can be used to calibrate the climate model outputs that accurately resemble the observed data during the calibration period. Then, the calibrated relationship between the model and observed data was applied to the validation period or future period.

### 2.3.3. Spatial Disaggregation

After bias-correcting the climate model, the long-term mean of the model was removed to obtain the anomalous field (by dividing the data by the daily long-term mean). The daily long-term mean was calculated using a 30-day running window centered on each day of the year. The anomalous field of the bias-corrected model data was then interpolated to a fine scale that matches that of the observation. Next, the interpolated anomalous field was scaled by the long-term mean of the observation to align the model's spatial feature with the spatial pattern of the observation. Finally, the downscaled climate model data were obtained.

### 2.4. BCSD Method Performance

Before applying the BCSD method to downscale climate models for climate change projection, we evaluated the performance of the method. We employed statistical indicators such as time-series correlation, pattern correlation (correlation of 2D map rearranged into 1D), and root mean square errors (RMSE) to evaluate the performance of the BCSD method. A detailed description of these statistical indicators can be found in [19]. Additionally, we also checked the performance in terms of climatological annual cycle pattern.

### 2.5. Climate Change Projection

After downscaling the climate models, we predicted the future climate patterns in the near-future period (2015–2045) and far-future period (2070–2100) under two climate change scenarios, using the downscaled climate model data. This research employed two Shared Socioeconomic Pathways scenarios (SSPs), namely SSP245 and SSP585. SSP245 is a climate change scenario that assumes moderate socioeconomic development and moderate climate change mitigation. The number 45 refers to the radiative forcing of 4.5 watts per square meter by the year 2100. SSP585 is the worst-case climate change scenario that assumes high greenhouse gas emissions and high temperatures. The number 85 refers to the radiative forcing of 8.5 watts per square meter by the end of the year 2100. The main objective of this study was to project the changes in mean precipitation and extreme rainfall events, using extreme climate indices.

### 2.6. Climate Change Indicators

In this study, we projected two types of precipitation change information over Cambodia after downscaling the coarse-resolution climate model to 10 km resolution. Firstly, we calculated the mean precipitation changes. Secondly, the CDD and CWD were calculated alongside the rx1day. These climate indicators are important for climate change projection linked to flood, drought, and rainfall pattern changes. The CDD is an indicator used for measuring drought severity and monitoring the changes in precipitation patterns over time [20]. The CWD is a climate index that measures the longest period of consecutive days with at least 1 mm of precipitation, as defined by the expert team on climate change detection and indices (ETCCDI) [21]. The rx1day is a climate index used for measuring the amount of precipitation that falls in a single day. The rx1day can be used to assess flood risk; monitor changes in precipitation patterns; and predict the likelihood of future flood events, such as flash floods or riverine floods [22].

Furthermore, the multi-model ensemble of each climate indicator was calculated. The process was simply done by dividing the downscaled climate model of the future time frame into two periods, which are the near future (2015–2045) and far future (2070–2100), and subtracting from the baseline period (1980–2014). Then, the median of the five climate models was calculated (multi-model ensemble) to conclude a more reasonable outcome of precipitation change projection rather than showing the individual results of climate model.

## 3. Results

### 3.1. Performance Evaluation of BCSD Method

Downscaling is very important in this study; therefore, we performed a cross-validation of the BCSD method on the historical data from CMIP6 models and compared it with observed data. In the evaluation process, we used the data from 1980 to 2014 and divided them into two time periods: calibration period (1980–2004) and validation period (2005–2014). The calibration is denoted as the baseline period, while the validation is denoted as the future period. Thus, the performance of BCSD relies more on the validation outcomes as the climate projection is done in future periods.

Figure 3 presents the improvement of the climate models after downscaling in terms of pattern correlation and RMSE of the climatological annual cycle map. Figure 3a shows the pattern correlation of the climatological annual cycle map of the raw model and the downscaled model with observation. It can be seen that the correlation of the downscaled model with observation improves drastically from January to December. Additionally, Figure 3b shows the RMSE of raw and downscaled models with observation. The RSME of the downscaled model with observation reduces significantly each month compared to the raw model. However, only in November and December, the RSME of the downscaled model slightly surpasses the raw model.

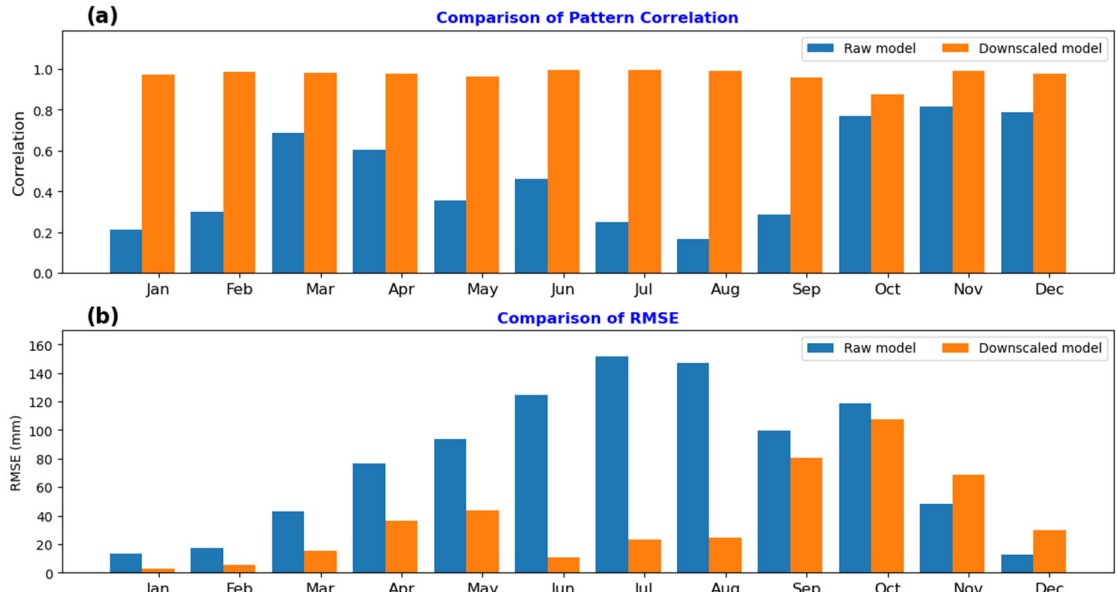

**Figure 3.** Pattern correlation (**a**) and RMSE of climatological annual cycle for the MPI-ESM1-2-HR model (**b**). Sample size for correlation calculation is 2500, which is the number of grid cells over Cambodia. Raw model data are bilinear interpolated to the same resolution as downscaled model data.

Figure 4 shows the correlation between the downscaled data and observed data. As seen on each map plot, the correlation of each climate model indicated a good relationship, with an average value of more than 0.7. Moreover, the highest correlation mostly occurs in the high-elevation area (northeastern part of the country) and coastal area (southwestern part).

Figure 5 illustrates the root mean square errors of downscaled and observed data. The highest error mostly happens in the coastal area (bottom left of the map plot), resulting from the orographic effect varying the rainfall amount. Similarly, there are slight biases over the high-elevation area (top right of the map plot), while the low-land area shows the smallest biases (central part of the country).

In summary, the results of cross-validation indicated the satisfactory performance of the BCSD method, which is suitable for application in Cambodia.

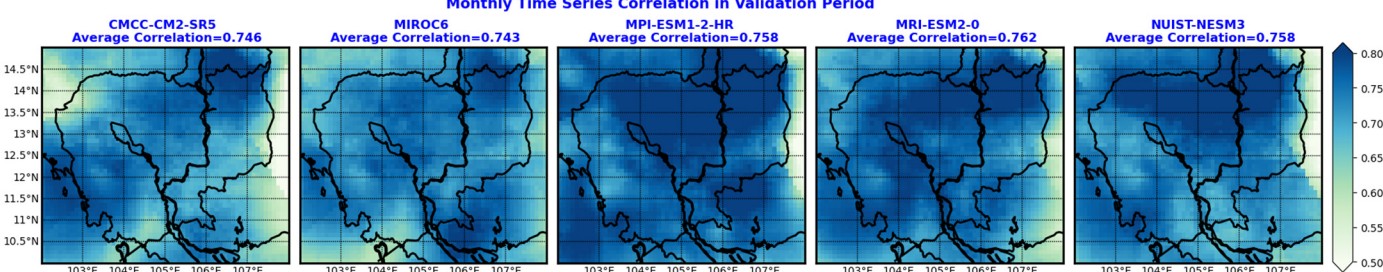

**Figure 4.** Correlation of monthly time series on each grid cell in validation period (2005–2014). Sample size for correlation calculation is 120, which is the number of months during the 10-year validation period.

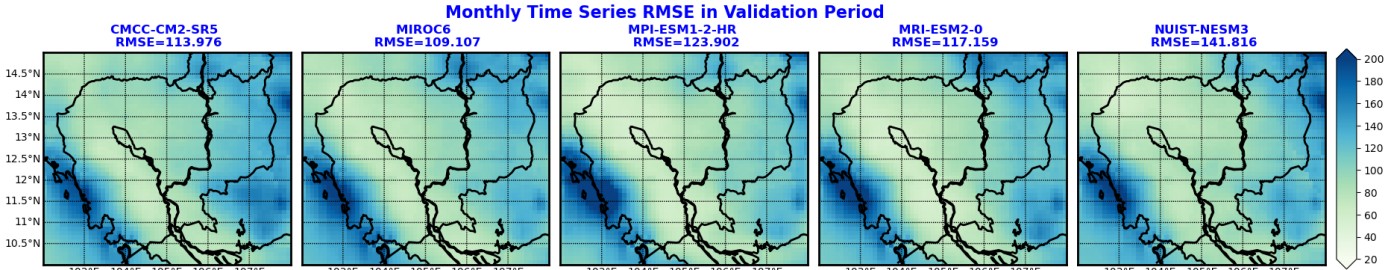

**Figure 5.** RMSE of monthly time series on each grid cell in validation period (2005–2014).

*3.2. Projection of Mean Precipitation*

After verifying the performance of the BCSD method, we proceeded to apply BCSD to the future climate model data and then used the outcome to predict future changes in precipitation. The future monthly long-term mean precipitation was calculated for each month of the year and then subtracted with the baseline period to figure out the percentage changes in the rainfall pattern. As Cambodia is highly influenced by the monsoon circulation, the seasonal match of rainfall pattern in the country is divided into dry season, pre-monsoon, full monsoon, and post-monsoon [14]. Thus, we illustrate the mean precipitation change for the following key months: dry monsoon (January), pre-monsoon (April), full monsoon (August), and post-monsoon (October). The projected change in mean annual rainfall is also analyzed. Furthermore, the variation in rainfall distribution is also due to the country's topographical conditions.

Figure 6 visualizes the projection of each climate model for mean precipitation changes under scenario SSP245 in the far future period. Different climate models indicated different rainfall change tendencies, and most models show a drastic increase in future rainfall. However, the result from MRI-ESM2-0 shows that, in three out of four months (except August), the rainfall tends to decrease significantly to a maximum of 40%. Even though each model produces enormous rainfall changes, the calculation of the multi-model ensemble shows that the percentage ranges from negative 20% to 40%, which follows the trend of CMCC-CM2-SR5, MRI-ESM2-0, and NUIST-NESM3; notably, the northwestern part would face the most drastic change. Furthermore, the annual rainfall changes can be seen increasing for all climate models, except for MRI-ESM2-0 which is decreasing by 10% for the majority part of the country. Similarly, the annual rainfall changes in the multi-model ensemble also align with the trend of the other four climate models. From an overall perspective, the dry season is getting drier, while the wet season is getting wetter, which could result in drought and flood events.

Figure 7 illustrates the change in mean precipitation under scenario SSP585 for far future period (2070–2100). The trend of precipitation changes under SSP585 shows that there is even more increase in rainfall than in scenario SSP245. For January, most climate models project a huge decline in rainfall, notably for MRI-ESM2-0, which is around 40%

decrease; the exception is MIROC6, which shows a positive percentage of up to over 40%. In addition, CMCC-CM2-SR5 and MRI-ESM2-0 also show a decreasing rainfall in April, covering the entire domain, while others show the opposite result. However, in August and October, most models show a consistently increasing rainfall trend in Cambodia. The multi-model ensemble is essential to conclude the overall changes. The multi-model ensemble for each month shows a strong decrease in rainfall in January (+10% to +40%) and a mixture of signals in April (−10% to +10%). The annual rainfall changes in each climate model follow the trend of the projection under scenario SSP245 (Figure 6) while only different in percentage. However, there is a slight difference in the CMCC-CM2-SR5 that contains a minority of decreasing rainfall in the southern part. The multi-model ensemble shows a slight increase in the annual rainfall over the whole country. In conclusion, the rainfall in Cambodia would likely decrease in the dry season and increase in the wet season, and it would slightly increase if aggregated to the annual amount.

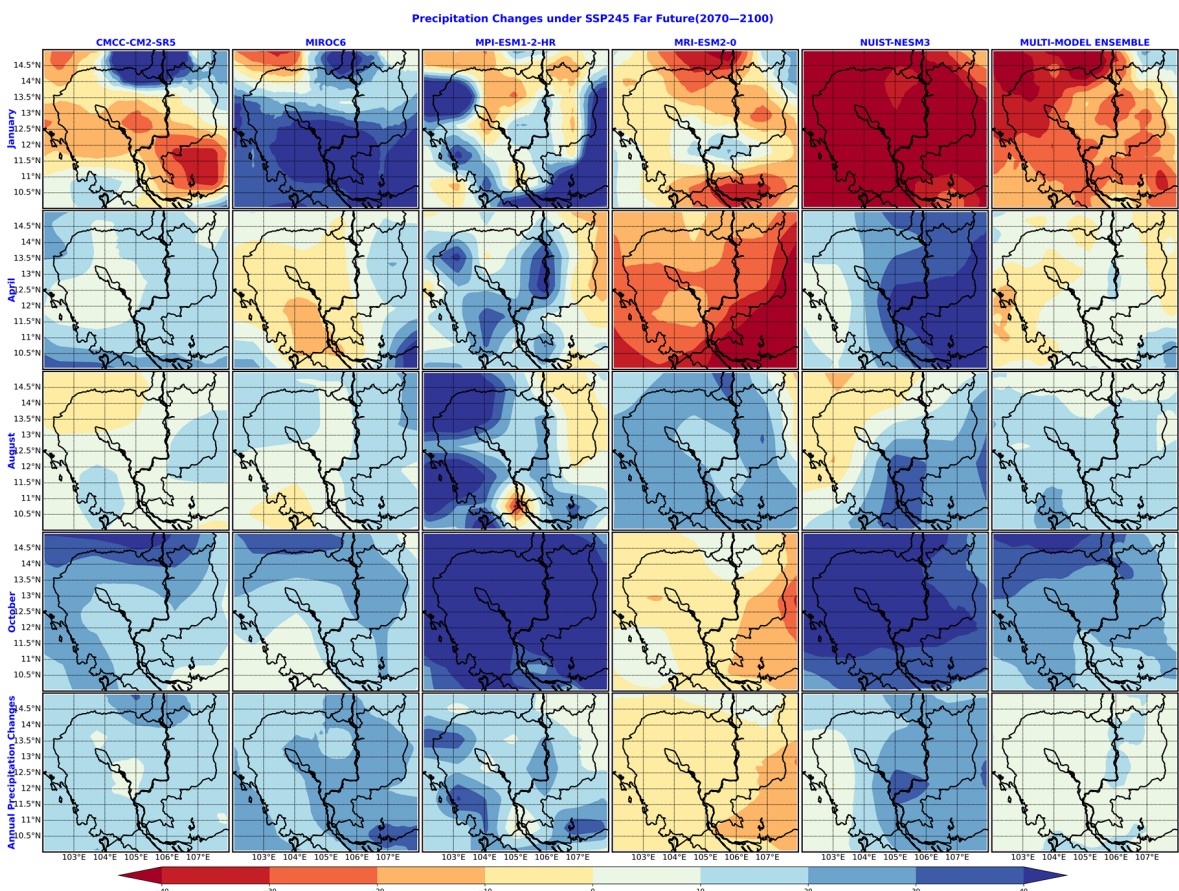

**Figure 6.** Percentage changes in climatological monthly precipitation for far future period (2070–2100) under SSP245 scenario. First-row panels for January, second-row panels for April, third row panels for August, and forth row panels for October. Note that the multi-model ensemble is the median among five models.

### 3.3. Projection of Consecutive Dry Days

The consecutive dry day (CDD) is a climate index used to measure the dry spell that is linked to a drought event which is determined by counting the number of days consecutively with the amount of less than a specific value of precipitation (1 mm in this study). We used the downscaled data of each climate model for the projection of the CDD under scenarios SSP245 and SSP585 and divided them into two time periods: near and far future periods. The information about the CDD can be used to spotlight the drought area and prepare an adaptation plan dealing with water shortage.

Figure 8 illustrates the CDD projection under SSP245. Most climate models for both time frames show that the CDD is decreasing. However, models like CMCC-CM2-SR5 and MRI-EMS2-0 show an increase in the CDD in the southern and southeastern parts of the country. The multi-model ensemble balances off the uneven percentage changes in the CDD among the five models. These consequences make the multi-model ensemble twist the outcome with a slight increase of around 10% covering a larger portion of the domain. However, some areas would have a slight decrease in the CDD, but mostly refer to the near future period, while it can hardly be detected in the far future period.

Figure 9 shows the projected changes in the CDD under scenario SSP585. For the near future period (first-row panels), almost all climate models project a decreasing percentage of the CDD, excluding CMCC-CM2-SR5, which shows an increasing trend. However, the multi-model ensemble indicates that the CDD would likely decrease up to 10% for most parts of Cambodia, while the CDD increases around 10% over small areas in the middle of the part of country, as well as near the border with Thailand, Lao PDR, and Vietnam. In the far future period (second-row panels), MPI-ESM1-2-HR and NUIST-NESM3 project both rising and dropping percentages of the CDD that range from minus 20% to 40%, while CMCC-CM2-SR and MRI-ESM2-0 indicate a stronger increase over the whole of Cambodia. Lastly, the multi-model ensemble signaled that the CDD will slightly increase to around 20%, especially over the coastal region (bottom right panel of Figure 8).

In summary, the projection of the CDD suggests an increase in the CDD over most parts of Cambodia, especially under the unmitigated world scenario SSP585 at the end of the century. This can be inferred as the potential threat for dry spells and drought in the country due to climate change impacts.

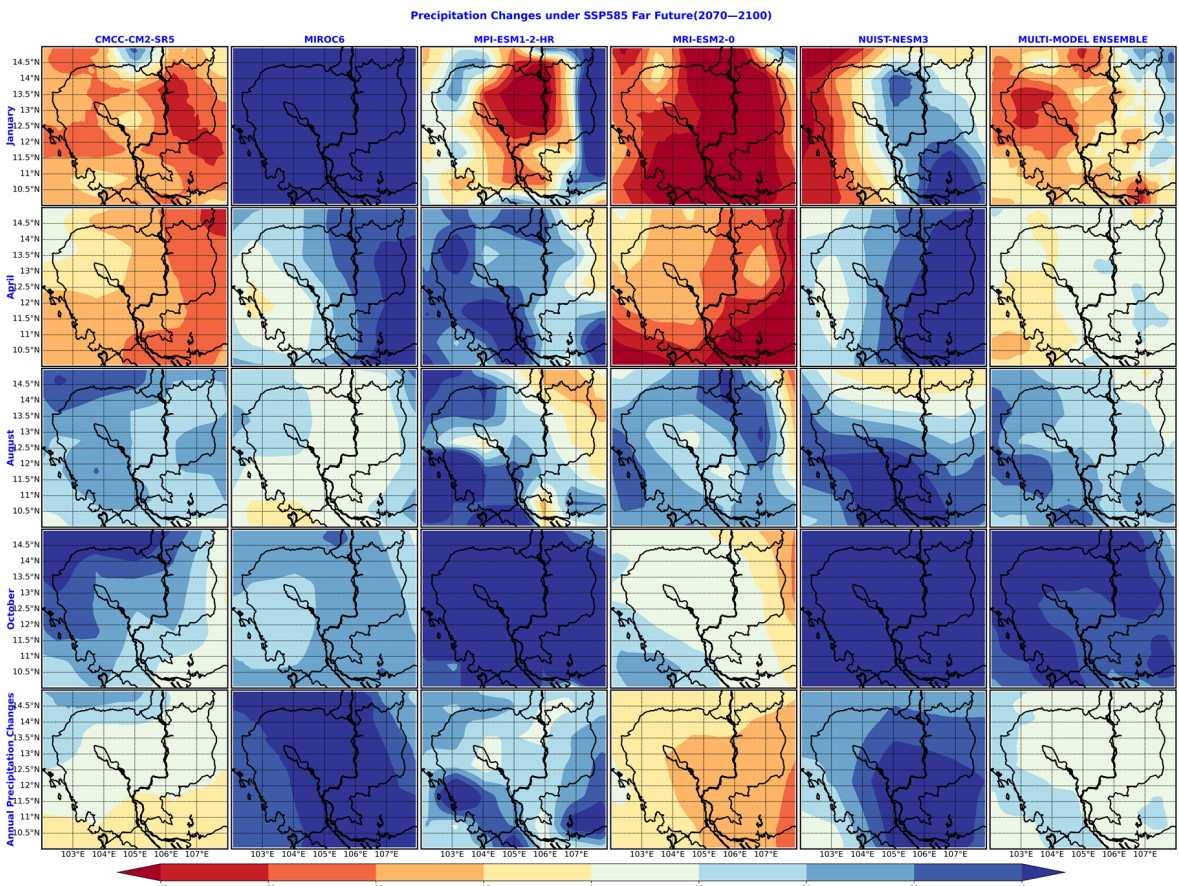

**Figure 7.** The same as Figure 6, for SSP585 scenario.

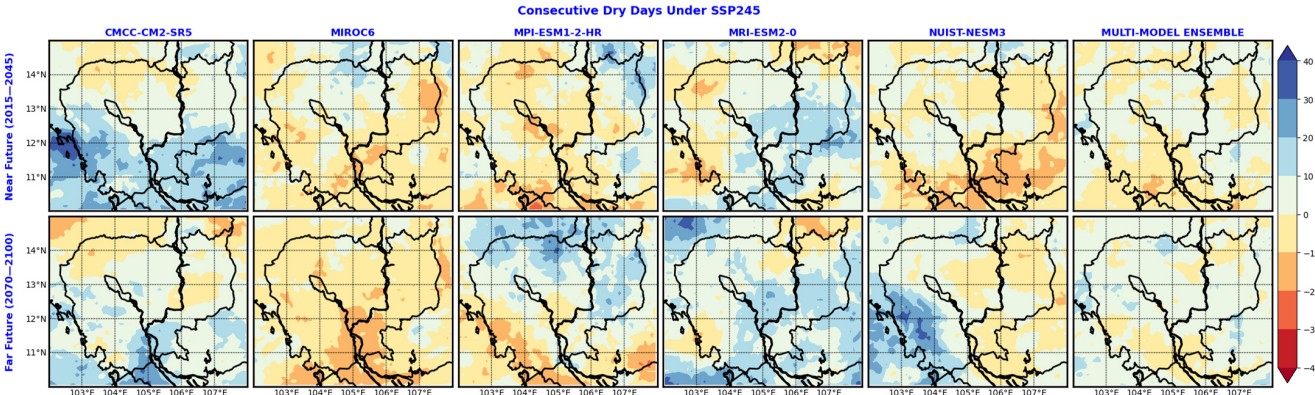

**Figure 8.** Percentage changes in consecutive dry day under scenario SSP245. First-row panels for near future (2015–2045), and second-row panels for far future (2070–2100).

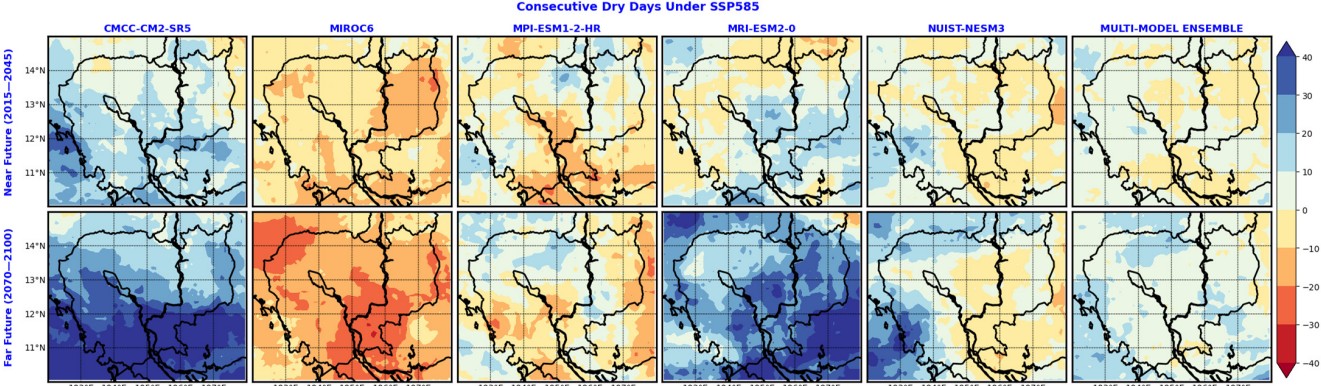

**Figure 9.** The same as Figure 8, for scenario SSP585.

*3.4. Projection of Consecutive Wet Day*

Just the opposite of the CDD, an expert team on climate change detection and indices (ETCCDI) defined the CWD as the climate index used for measuring the longest consecutive days with at least 1 mm of precipitation [21]. The CWD is calculated by counting the number of consecutive days with more than a specified amount of rainfall (1 mm in this study). Regarding the information on the CWD, it can be used to project flooding or indicate areas that might be affected by prolonged periods of rainfall. Moreover, if the projection of the CWD is decreased, it would likely show the possibility of dry spell.

Figure 10 illustrates the projection of the CWD under scenario SSP245. For the near future period (first-row panels), CMCC-CM2-SR5 and NUIST-NESM3 show that the whole domain would likely experience a decrease in CWD, while the other three models indicate a similar pattern that contains both increasing and decreasing CWD. However, the multi-model ensemble indicates a mixture of decreases (diagonally from southwest to northeastern parts) and increases (northwestern and southeastern parts) in the CWD in Cambodia. For the far future projection (second-row panels), most of the models show positive percentage changes in CWD, excluding NUIST-NESM3, which shows negative percentage changes. Therefore, the multi-model ensemble shows the increasing CWD over the whole domain of Cambodia.

Figure 11 indicates the projection of the CWD under scenario SSP585. In the near future period (first-row panels), the CMCC-CM2-SR5 and NUIST-NESM3 models produced almost identical patterns of decreasing CWD up to around 20%. Meanwhile, MIROC6 shows a significant increase in CWD with few declining spots. Likewise, the MPI-ESM1-2-HR and MRI-ESM2-0 shared a similar pattern with a rising percentage in the western parts of the country and a descending percentage in the eastern part. However, the multi-model

ensemble suggests an increasing trend of the CWD with merely positive and negative 10% scrambling over the entire domain. For the far future period (second-row panels), all models except CMCC-CM2-SR5 project a similar spatial pattern of CWD changes: an increase in the western part and a strong decrease in the eastern part. Therefore, the multi-model ensemble also has a similar pattern to the four models (increase in the western part and strong decrease in the eastern part), even though CMCC-CM2-SR5 has large negative percentage changes.

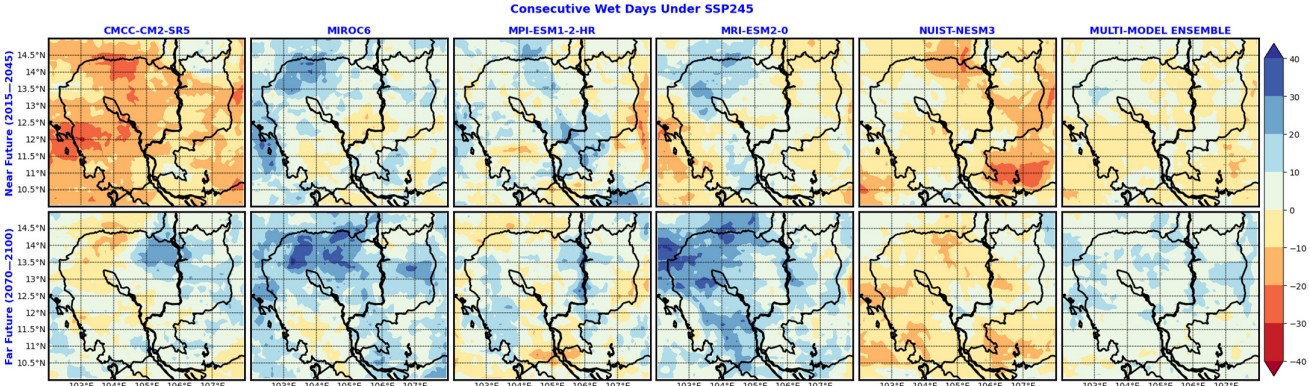

**Figure 10.** Percentage changes in the consecutive wet days under scenario SSP245. First-row panels for near future (2015–2045), and second-row panels for far future (2070–2100).

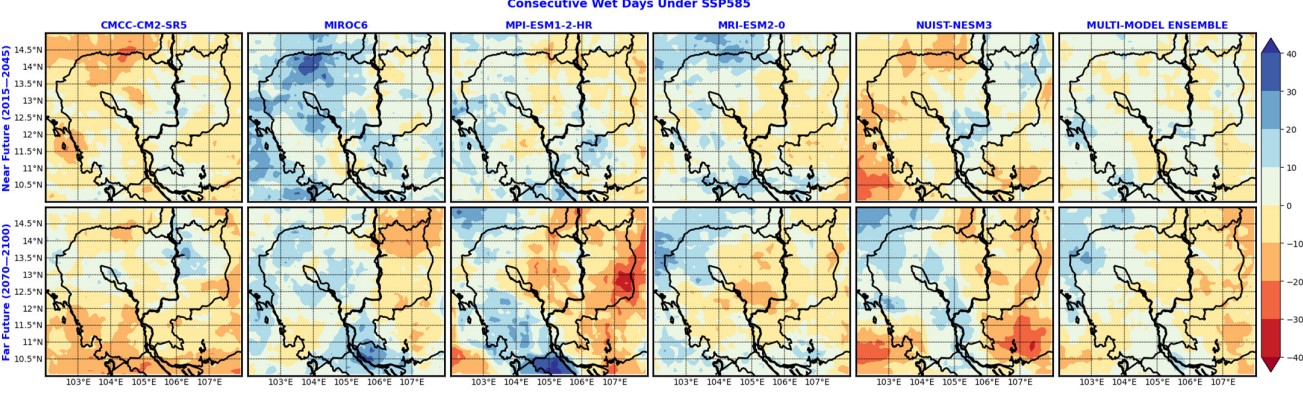

**Figure 11.** The same as Figure 10, for scenario SSP585.

In total, the projection of the CWD suggests an increase in the CWD under the moderate scenario (SSP245) and a strong decrease in the eastern part of the country under the unmitigated world scenario (SSP585). This can be inferred that the wet spell would happen under the moderate scenario of climate change, but it would be the opposite under the unmitigated world scenario (SSP585), as the number of wet days would be reduced and then replaced with a dry spell that is consistent with the CDD projection (Figure 8).

### 3.5. Projection of Maximum 1-Day Precipitation

The maximum 1-day precipitation (rx1day) is a climate index that measures the highest amount of rainfall accumulated during a day record (24 h). The projection of the rx1day can be essential for preparing mitigation and adaptation responding to flooding events such as flash floods or riverine floods over specific regions. The information on rx1day is yet limited; therefore, before using it for any decision making, users must be aware of the nature of the model bias and take advantage of multi-model information.

Figure 12 presents percentage changes in the rx1day under scenario SSP245. In the near future period (first-row panels), the projection shows that rx1day would likely increase from 10% to more than 40%, especially for MIROC6, MPI-ESM1-2-HR, and NUIST-NESM3, while

the other two models indicate a small decrease in the northern part of Cambodia. Even though some models have some downfalls of rx1day percentage, the multi-model ensemble shows positive percentage changes in the rx1day from around 10% to 30%. Most notable is the increasing pattern of 20% spread over the central part from northwest to southeast of the country. For the far future period (second-row panels), the rx1day projections of all models show that percentage changes in the rx1day will be soaring from 10% to more than 40%, especially the MPI-ESM1-2-HR model that has the most remarkable change up to over 40% spread over Cambodia. However, the MRI-ESM2-0 shows a detectable rx1day decrease for some regions, mostly the northern part of the country. Lastly, the multi-model ensemble shows increase with a large percentage of variability spreading all over Cambodia.

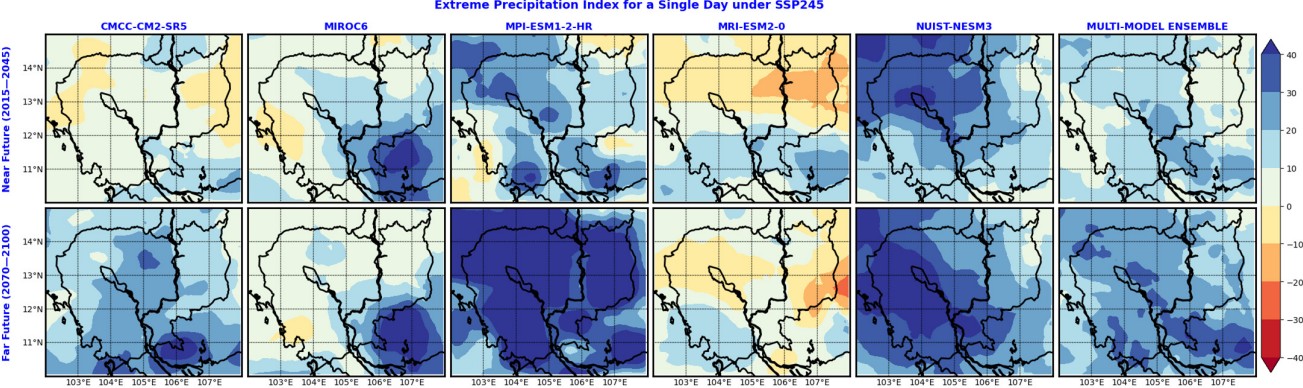

**Figure 12.** Percentage change in rx1day under scenario SSP245. First-row panels for near future (2015–2045), and second-row panels for far future (2070–2100).

Figure 13 illustrates the projection of the rx1day under scenario SSP585. The projection in the near future period (first-row panels) shows that MPI-ESM1-2-HR and NUIST-NESM3 have the most significant increase in the rx1day over the whole country compared to the other three models, thus indicating a mixture of increases and decreases in the rx1day over the country. Meanwhile, the result of the multi-model ensemble concludes that the rx1day would likely increase from around 10% to 30%. For the far future period (second-row panels), each model projects a similar trend, showing that the rx1day would be rising and covering the entire domain, except for some areas of the northern part of the country, from MIROC6 and MRI-ESM2-0 models, with decreasing values from around 10% to 20%. Moreover, CMCC-CM2-SR5, MPI-ESM1-2HR, and NUIST-NESM3 show strong percentage changes in the rx1day to more than 40% over Cambodia. Lastly, the multi-model ensemble sums up the result with an increasing percentage ranging from 30% to more than 40% (bottom right panel of Figure 12).

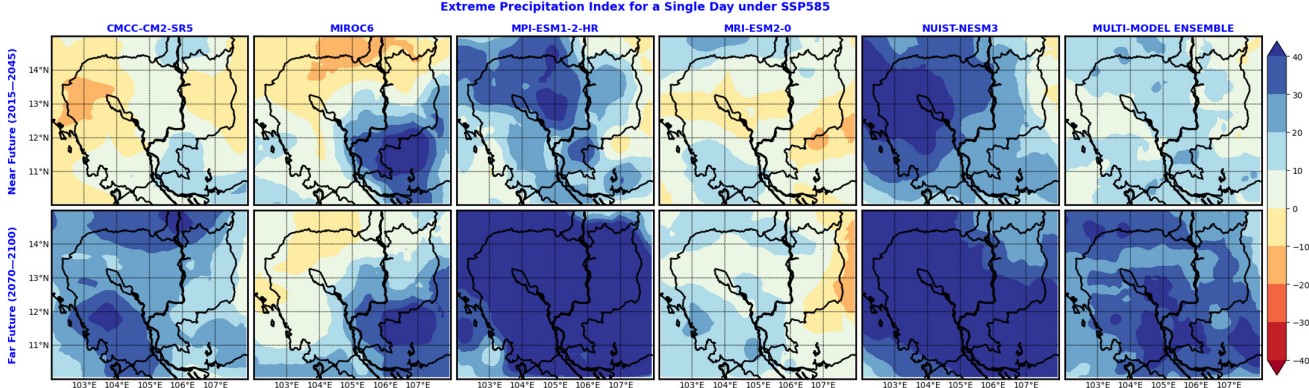

**Figure 13.** The same as Figure 12, for scenario SSP585.

In summary, the project of rx1day suggests the increase in the rx1day over most parts of Cambodia, especially under the unmitigated world scenario SSP585 at the end of the century. This can be inferred as the potential threat of extreme rainfall triggering floods in the country due to climate change impact.

## 4. Discussion

The statistical downscaling method, BCSD, has been applied widely in different parts of the world [9,18,23]; especially the application by the NASA team to produce the NASA Earth Exchange Global Daily Downscaled Projections (NEX-GDDP-CMIP6) [24]. However, this product still has a rather coarse resolution (0.25 degrees) and is still limited to exploring climate change information in small countries, like Cambodia. In this study, we explored the application of the BCSD method in Cambodia by using a higher-resolution observation dataset (MSWEP) that we could use to downscale the precipitation up to 0.1 degrees over Cambodia. We also proved the good performance of this method in regard to capturing the spatial features of orographic precipitation in Cambodia (Figures 3–5). The downscaled data can capture the month-to-month variation very well, especially in the areas with high amounts of orographic rainfall (coastal area and northeast mountain hill area) (Figure 4), but large biases were noticed in those areas instead (Figure 5). This may be caused by the disaggregation procedure of the anomalous bias-corrected field of the climate model output in the BCSD.

This is the first-ever application of the BCSD method to downscale climate model output over Cambodia, in which the up-to-date climate model data of CMIP6 were used in this study. Moreover, most of the previous studies focused on only the changes in mean precipitation [11–13]; limited studies focus on the changes in extreme precipitation, which is very pronounced under the impact of climate change. Therefore, a study related to the projected changes in extreme precipitation in Cambodia based on CMIP6 models is required to fill in this gap.

The results of the projection of the mean precipitation change, CDD, CWD, and rx1day based on the finer-scale downscaling data in the current study conclude that the future climate of Cambodia would likely be getting drier during the dry season and wetter during the rainy season. The current study also suggests the potential threats of more dry spell events related to drought and more heavy rainfall events related to floods in most parts of Cambodia as the impact of climate change. This result is consistent with the law of atmospheric physics that the warm air can store more water vapor for a longer time (causing dry spell). However, when those large amounts of water vapor become rain droplets, it would cause stronger rainfall intensity, triggering flood. Moreover, several previous research studies indicate the same trend as our results. Climate projection over Southeast Asia using the regional climate model done by [25] showed that the studied area would be experiencing drier conditions with more heavy rainfall events. Additionally, our finding shows that future climate change aligns with the IPCC, with a rise in rainfall in the rainy season and a decline in rainfall in the dry season. Our projection on extreme events also indicates that Cambodia is vulnerable to drought and flooding, the same as the research carried out by [26]. Another study also shows that the flow in several streamlines in Cambodia is predicted to decrease in the future compared to the baseline period due to the lack of rainfall in the dry season [27]. Likewise, the projection of precipitation in the Siem Reap province of Cambodia also suggested that rainfall will be decrease in the dry season [13]. The current study also captures the same trend. The result of this study could add up the confidence of projection for the future change in rainfall patterns in Cambodia based on the newly updated climate modeling activity, CMIP6.

## 5. Conclusions

In this study, we explored the possibility of applying a statistical downscaling method, BCSD, in downscaling the climate model output for the climate change study in Cambodia. Based on the cross-validation and the statistical indicators (correlation, RMSE, and pattern

correlation), we proved the good performance of this method by capturing the spatial feature of orographic precipitation in Cambodia (Figures 3–5). This indicates that the BCSD method is suitable for application in Cambodia.

The BCSD method was then applied to downscale precipitation variables from five CMIP6 climate model outputs (Table 1) for a historical run and two climate change scenarios: SSP245 and SSP585. The downscaled data were employed to project the precipitation pattern change over Cambodia in terms of mean precipitation and three extreme climate indices that link to dry spells and flood events. The results suggest that the future climate of Cambodia would likely be getting drier during the dry season and wetter during the rainy season. The current study also suggests the potential threats of more dry spell events related to drought and more heavy rainfall events related to floods in most parts of Cambodia as a result of the impact of climate change.

The application of the BCSD method is a key to assessing future climate information with a reasonable projection on climate indices. From the current study, we can capture images of future climate behavior depending on the climate change scenarios. On top of that, using the BCSD method with the CMIP6 dataset can be considered a path for the next generation of researchers to continue projecting future climate with minimal resources that can be implemented for decision making in regard to climate change mitigation and adaptation. Even though there are a significant number of limitations of BCSD method, we managed to produce results that seem to follow the track of many previous studies, making the current study a success in this regard and parallel with the hypothesis we made before conducting this research.

**Author Contributions:** Conceptualization, R.C., L.S. and S.D.; methodology, R.C. and S.D.; software, R.C. and S.D.; validation, R.C. and S.D.; formal analysis, R.C. and S.D.; investigation, R.C. and L.S.; resources, R.C. and L.S.; data curation, R.C. and S.D.; writing—original draft preparation, S.D.; writing—review and editing, R.C. and L.S.; visualization, R.C. and S.D.; supervision, R.C. and L.S.; project administration, L.S.; funding acquisition, L.S. and R.C. All authors have read and agreed to the published version of the manuscript.

**Funding:** The work was funded by the Cambodia Higher Education Improvement Project (HEIP, Credit No. 6221-KH) and the Cambodia Climate Change Alliance phase 3 (CCCA3).

**Data Availability Statement:** In this the data were obtained from the freely available data portal including Coupled Model Intercomparison Project Phase 6 (CMIP6) data portal (link: https://esgf-node.llnl.gov/search/cmip6/) and Multi-Source Weighted-Ensemble Precipitation (MSWEP) for the free precipitation dataset (link: https://www.gloh2o.org/mswep/).

**Acknowledgments:** The work was fully supported by the Cambodia Higher Education Improvement Project (HEIP, Credit No. 6221-KH) and the Cambodia Climate Change Alliance phase 3 (CCCA3). Moreover, I would like to express my sincere thanks to open data sources such as Coupled Model Intercomparison Project Phase 6 (CMIP6) data portal and Multi-Source Weighted-Ensemble Precipitation (MSWEP) for the free precipitation dataset.

**Conflicts of Interest:** The authors declare no conflict of interest.

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
