# Peer review of "Precipitation Projection in Cambodia Using Statistically Downscaled CMIP6 Models"

_climate, doi:10.3390/cli11120245_

Round 1
Reviewer 1 Report
Comments and Suggestions for Authors
Comments on manuscript entitled “ Precipitation Projection in Cambodia Using Statistically-2 Downscaled CMIP6 Models”, by DUONG et al.
The manuscript deals with applying a statistical downscaling method, named BCSD, to downscale the climate model output over Cambodia. Then they applied the downscaled data to project the future condition of precipitation pattern and extreme over the study region. While the manuscript is well-structured and results are well presented, there are still a few points need to be address before its publication in Water journal.
The most critical point is the novelty of this research: In introduction section, the authors should explicitly indicate the innovations of their study.
Introduction: More relevant research on climate change in Combodia should be used and cited.
Section 2-2-3: More details on Spatial disaggregation should be provided, preferably supported by a flowchart/diagram.
Figures 3-5: what climatological variable these figures are for? Precipitation? Temperature?
Another important point is the discussion section: More similar studies in Combodia should be used in this section to compare the outcomes of this research with previous ones.
Comments on the Quality of English Language
There are some issues need to be addressed. The manuscript should be revised to improve the grammar points.
Author Response
We would like thank the constructive comment from the reviewer. Based on your comment, we made a substantial revision to the manuscript and we found a lot of improvement to the current version of the manuscript. We address the point-by-point comments in the attached response letter

Reviewer 2 Report
Comments and Suggestions for Authors
In this study, the BSCD downscaling method was used to downscaling precipitation data from model output of CMIP6, and future precipitation pattern and extreme in Cambodia were also evaluated using projected data and several climate indicators. The analysis and writing are generally fine. However, there are several issues could be addressed, and the quality of the manuscript could be improved. My comments are as follows:
1. Figure 2 and 4: It suggested to add the size of samples in the figure captions, since the significance of correlation coefficient is highly related to the sample size.
2. Figure 4: The titles of each sub-plot should be “Average RMSE value” rather than the “Average Correlation”.
3. As shown in Figure 2 and 4, the downscaled precipitation data showed opposite performances in the high elevation regions (south-west areas) as revealed by correlation coefficient and RMSEs. More explanations about this phenomenon should be added.
4. Line 168: “Figure 4” seems to be “Figure 3”, and the authors should carefully check the orders of figures, especially for Figures 2-4.
5. It is suggested to add a brief introduction about the study area in the section “Materials and Methods”, as not all the readers are familiar with the natural conditions (such as elevation distribution, monsoon, climate type and pattern, etc., which are mentioned in the manuscript) of Cambodia.
6. As shown in Figures 7 and 8, the multi-model ensemble showed a quite different spatial pattern to that of other models. More technical details about how is the “multi-model ensemble” calculated should be added in the section “Materials and Methods”. And more explanations about the difference between the ensemble with other model outputs should be added in Section 3.3.
7. Section 3.2: In addition to the precipitation projections in those typical months that has been presented (Figures 5 and 6), the projections of annual mean (or total) precipitation is suggested to be added in the Figures. This could help take a more comprehensive and intuitive picture of the future precipitation.
8. More details or references about the climate change indicators should be added in Section 2.5, instead of being provided in the results section (such as Lines 216-218, Lines 247-253, and so on).
Comments on the Quality of English LanguageMinor editing of English language required.
Author Response
We would like to thank the constructive comment from the reviewer. Based on your comment, we made a substantial revision to the manuscript and we found a lot of improvement to the current version of the manuscript. We address the point-by-point comments in the attached response letter.

Reviewer 3 Report
Comments and Suggestions for Authors
This manuscript is well-organized. After the necessary revisions, I feel this manuscript can be approved.
The introduction must be expanded out, especially by include a lot more references. Aside from the six sources, no one else has done research on this topic? The Discussion section is also quite brief.
There is no section that contains a physiographic description of the study area.
What underlies the selection of these five climate models?
It is advised to utilize the median when calculating the ensemble mean. This will aid in assigning weights to individual models when, for example, one of them is significantly different.
Author Response

(The authors gave the same response as above.)

Round 2
Reviewer 3 Report
Comments and Suggestions for Authors
Thanks for some improvements to the manuscript. Your responses are very sparse. I am not sure that this article will have much resonance in the scientific community, but I wish the authors success, subject to the editor's support.
Comments on the Quality of English LanguageMinor editing of English language required
Author Response
We thank you for your comments and we make additional revisions to our manuscript. We added more description to the introduction section and cited more research related to climate downscaling. We also make careful revisions to English writing with the assistance from Grammarly and online English checking in Microsoft.